# A novel dual-branch network for comprehensive spatiotemporal information integration for EEG-based epileptic seizure detection

**Xiaobing Deng**[ID]*

School of Computer Information Engineering, Nanchang Institute of Technology, Nanchang, Jiangxi, China

* guoguo10201@163.com

**Data availability statement:** The data is publicly available. The data underlying the results presented in the study are available from (https://physionet.org/content/chbmit/1.0.0/).

## Abstract

Epilepsy is a neurological disorder characterized by recurrent seizures caused by abnormal brain activity, which can severely affects people's normal lives. To improve the lives of these patients, it is necessary to develop accurate methods to predict seizures. Electroencephalography (EEG), as a non-invasive and real-time technique, is crucial for the early diagnosis of epileptic seizures by monitoring abnormal brain activity associated with seizures. Deep learning EEG-based detection methods have made significant progress, but still face challenges such as the underutilization of spatial relationships, inter-individual physiological variability, and sequence intricacies. To tackle these challenges, we introduce the Dual-Branch Deepwalk-Transformer Spatiotemporal Fusion Network (Deepwalk-TS), which effectively integrates spatiotemporal information from EEG signals to enable accurate and reliable epilepsy diagnosis. Specifically, the Spatio-branch introduces an adaptive multi-channel deepwalk-based graph framework for capturing intricate relationships within EEG channels. Furthermore, we develop a Guided-CNN Transformer branch to optimize the utilization of temporal sequence features. The novel dual-branch networks co-optimize features and achieve mutual gains through fusion strategies. The results of extensive experiments demonstrate that our method achieves the state-of-the-art performance in multiple datasets, such as achieving 99.54% accuracy, 99.07% sensitivity and 98.87% specificity. This shows that the Deepwalk-TS model achieved accurate epilepsy detection while analyzing the spatiotemporal relationship between EEG and seizures. The method further offers an optimized solution for addressing health issues related to seizure diagnosis.

## Introduction

Epilepsy, a chronic neurological disorder that affects approximately 50 million people worldwide, ranks second as the second largest neurosystemic disease globally. Epilepsy seizures are marked by sudden, transient, and repetitive, often accompanied by uncontrolled muscletwitches and short-term loss of consciousness [1–3]. Therefore, early diagnosis of seizures

**Funding:** This research was funded by the project "Design and Implementation of Urban Economic Intelligent Analysis and Management Platform (GJJ2202706)". The project host is Xiaobing Deng. The funders participated in a series of work in this study, including algorithm design, experimental validation, and preparation of manuscripts and publication.

**Competing interests:** The authors have declared that no competing interests exist.

is important for timely intervention, reducing the risk of irreversible brain damage, and improving the overall quality of life for patients [4–6]. Advanced medical imaging techniques typically detect epilepsy by identifying lesion information, but cannot capture ongoing seizures in the time domain. Therefore, to facilitate timely epilepsy diagnosis, Electroencephalogram (EEG), as a reliable method to capture brain electrical activity, has become a standard diagnostic technique in current clinical practices for epilepsy [7,8].

The computer-aided diagnosis of epilepsy based on EEG signals (EEGs) involves the automated screening of epileptic signals, representing a crucial step in achieving an accurate and efficient diagnosis. Expertise-dependent visual observations suffer from subjectivity and time-intensive analysis due to high-frequency acquisition of EEGs, leading to potential inaccuracies, misdiagnoses, and missed diagnoses [7,9]. Therefore, accurately decoding EEGs is crucial for assisting medical diagnosis, helping neurologists with treatment, and reducing risks in engineering.

The field of deep learning-based EEG seizure detection has attracted extensive research. However, existing methods do not fully consider the inherent characteristics of EEGs, such as multi-channel and temporal sequencing, resulting in insufficient feature extraction and low identification accuracy. The main challenges are as follows. (1) Underutilization of spatial relationships among EEGs across channels, impeding the conversion of raw EEG data into a structured graph. (2) The significant inter-individual physiological variability poses obstacles to deploying automated seizure detection, particularly when applied to unfamiliar patients. (3) The high complexity of EEGs encompasses aspects such as long duration, information redundancy, and a low signal-to-noise ratio due to various noise sources. (4) The extraction and fusion of features present challenges in handling long time series and integrating features in the frequency and time domain.

Therefore, to tackle the challenges, previous studies have explored the extensive applications of traditional feature-based machine learning algorithms (ML) [10] and deep learning algorithms (DL) [11,12] for automatic epilepsy detection. Initially, ML dominated epilepsy detection, but relying on manually designed features made it challenging to capture discriminative features. Recently, DL technology has been widely applied, automatically extracting meaningful features from EEGs in an end-to-end manner and enhancing accuracy. DL-based models are always categorized into Convolutional Neural Networks (CNN)-based [11], Long Short-Term Memory networks (LSTM)-based, Graph Convolutional Networks (GCN)-based [13], and hybrid models [14]. Researchers have proposed alternatives such as 3D CNNs [15] and multi-channel CNN networks [16]. Modified CNNs [17] exhibit discriminative feature learning in EEG classification. However, the CNN-based method struggles with long-term dependencies in time series analysis. LSTM-based methods have been introduced to capture temporal features [18], but always overlook graph structure information in EEGs. Hybrid models, such as CNN+LSTM, have been introduced in various studies to combine the strengths of different architectures and overcome the limitations of individual models.

It is well known that brain structural connectivity reflects the natural links between different regions, making it a key aspect of EEG analysis. The correlation between EEG data and epilepsy detection tasks can involve brain topology or nodal features. Therefore, GCN-based methods are widely used for modeling brain connectivity, as they leverage node characteristics and graph structure to capture complex relationships [19,20]. However, their effectiveness can be affected by poor graph connectivity and high node complexity. DeepWalk is a graph-based machine learning algorithm that learns latent representations of graph vertices by performing random walks [21]. Compared to GCN methods, it offers greater scalability, simplicity, and robustness to the quality of the graph structure.

To address these challenges, we introduce an innovative approach to seizure detection that incorporates both the temporal and spatial aspects of EEGs for the first time. The method, named Dual-Branch Deepwalk-Transformer Spatiotemporal Fusion Network (Deepwalk-TS), consists of two key components. Firstly, to fully extract spatial information between brain channels, we introduce a novel DeepWalk method to reconstruct the node features and topological structures of a brain channel graph. Then, we utilize GCN to extract features and enhance the model's ability to learn complex relationships. This approach allows for dynamically adjustable random walks, which is well-suited for multi-channel EEG seizure detection. Secondly, we propose a Guided-CNN Transformer branch to address the underutilization of temporal information in EEGs. It adopts CNN and a self-attention mechanism as a guide to enhance Transformer architectures, maximizing the extraction of crucial temporal features. The proposed dual-branch structure collaborates and enhances each other to achieve end-to-end spatiotemporal feature extraction of EEGs, thereby optimizing the performance and efficiency of epilepsy detection. Our contributions are as follows.

- Based on the characteristics of brain EEGs, we designed an end-to-end dual-branch network (Deepwalk-TS) specifically for spatiotemporal EEGs extraction and epileptic seizure detection.
- We propose a novel spatial adaptive multi-channel graph construction framework based on deepwalk. DeepWalk fully leverages the stochastic nature of random walks to efficiently extract relevant features between brain channels.
- We further introduce the Guided-CNN transformer branch for learning attention weights to improve the long-time consistency of EEGs. The outputs of these two branches are then fused using a joint optimization strategy.
- We conducted extensive experiments across multiple datasets and provided a detailed analysis of each branch. The results show that the Deepwalk-TS method achieves state-of-the-art performance, significantly improving the accuracy of epilepsy detection.

The rest of this paper is organized as follows: The Related Work reviews existing research on epileptic seizure detection, focusing on various machine learning and deep learning approaches. Methodology Section describes the proposed Deepwalk-TS model, explaining its architecture, key components, and spatiotemporal features. The Experiments present the datasets used, performance metrics, and comparison with recent studies. The Discussion section analyzes the implications of the results, limitations, and future directions. Finally, the Conclusion section summarizes the main findings of the study and outlines the potential impact of the proposed approach in real-world epilepsy detection applications.

## Related work

### Machine-learning techniques for epileptic seizure detection

Traditional epileptic seizure detection methods primarily rely on manual feature extraction and machine learning models, such as Support Vector Machines (SVM) and k-Nearest Neighbors (KNN) [22]. These methods often depend on fine preprocessing and feature engineering of EEGs. For instance, Sun used Discrete Wavelet Transform (DWT) and uniform one-dimensional binary patterns to extract texture features from EEG recordings [23]. Al-Hadeethi et al. achieved good results with the AB-LS-SVM model by reducing EEG dimensionality, extracting statistical features, and selecting key features [24]. Vicnesh et al. classified various types of epilepsy by analyzing nonlinear EEG features and organizing them into

a decision tree [25]. However, the transition of EEGs from non-ictal to ictal states is complex and dynamic, with significant differences in EEGs between patients and even between seizures in the same patient. Thus, accurately extracting seizure-related features remains a significant challenge.

### Deep-learning methods for epileptic seizure detection

ML has shown significant efficacy in detecting and recognizing abnormal behaviors in individuals, thus the need for automated detection and identification of these behaviors. Recently, the application of DL models for seizure detection has increased, especially with the use of CNN [11,26], GCN [13], and LSTM [18]. Emami et al. segmented filtered EEGs into one-second segments, converted them into images, and classified each image using CNN to distinguish between ictal and non-ictal states [27]. Hu et al. introduced an innovative approach using a deep bidirectional LSTM (BiLSTM) network for seizure detection [18]. Roy developed a multi-scale feature CNN with adaptive transfer learning for EEG motor imagery classification [28]. It improves accuracy by addressing inter-subject variability and combining convolution scales with transfer learning.

### Recent advances in epileptic seizure detection

While CNNs have demonstrated high performance in various studies, they often struggle with generalizing to new data or dealing with shifts in datasets. This limitation has prompted recent studies to explore transformer-based networks. For instance, Pan et al. proposed a transformer-based model for epilepsy detection, addressing sample imbalance by oversampling the minority class [29]. Shu et al. introduced EpilepsyNet [30], a transformer-based network that capitalizes on the strengths of transformers to enhance seizure detection accuracy.

In parallel, the success of GCNs in computer vision tasks, such as visual detection [31], disease prediction [32], and image segmentation, has inspired researchers to apply GCNs to epilepsy detection. GCNs excel at capturing complex relationships and structured data, making them highly effective for modeling the spatiotemporal and long-term dependencies in EEGs. Moreover, recent research has explored combining GCNs with transformers, further improving their performance in seizure detection tasks. Hybrid architectures that merge GCNs and transformers, like the GCN-Transformer models, have also shown significant promise in handling multimodal information.

## Materials and methods

In this section, we will provide a detailed introduction to the proposed Deepwalk-TS method for epilepsy detection. The method comprises two key branches, including the spatial graph reconstruction and the temporal sequence feature extraction. The former introduces a random Deepwalk-based GCN network to enhance the topological structures and node features. The latter Transformer is designed for temporal sequence extraction from the original EEGs. The detection model utilizes a joint loss optimization strategy to ensure mutual enhancement between the two branches. Fig 1 illustrates the overall Deepwalk-TS model framework, including the input of the initial graph structure and EEGs, the network details of the dual-branch spatiotemporal model, and the seizure prediction module and results. The model takes the raw EEGs as input, collected from the International 10-20 EEG electrode system. The main part of the model consists of the upper and lower sections of Fig 1. The upper figure shows the DeepWalk-based GCN used for reconstructing the channel graph module, while the lower figure displays the Guided-CNN Transformer module used for sequence feature

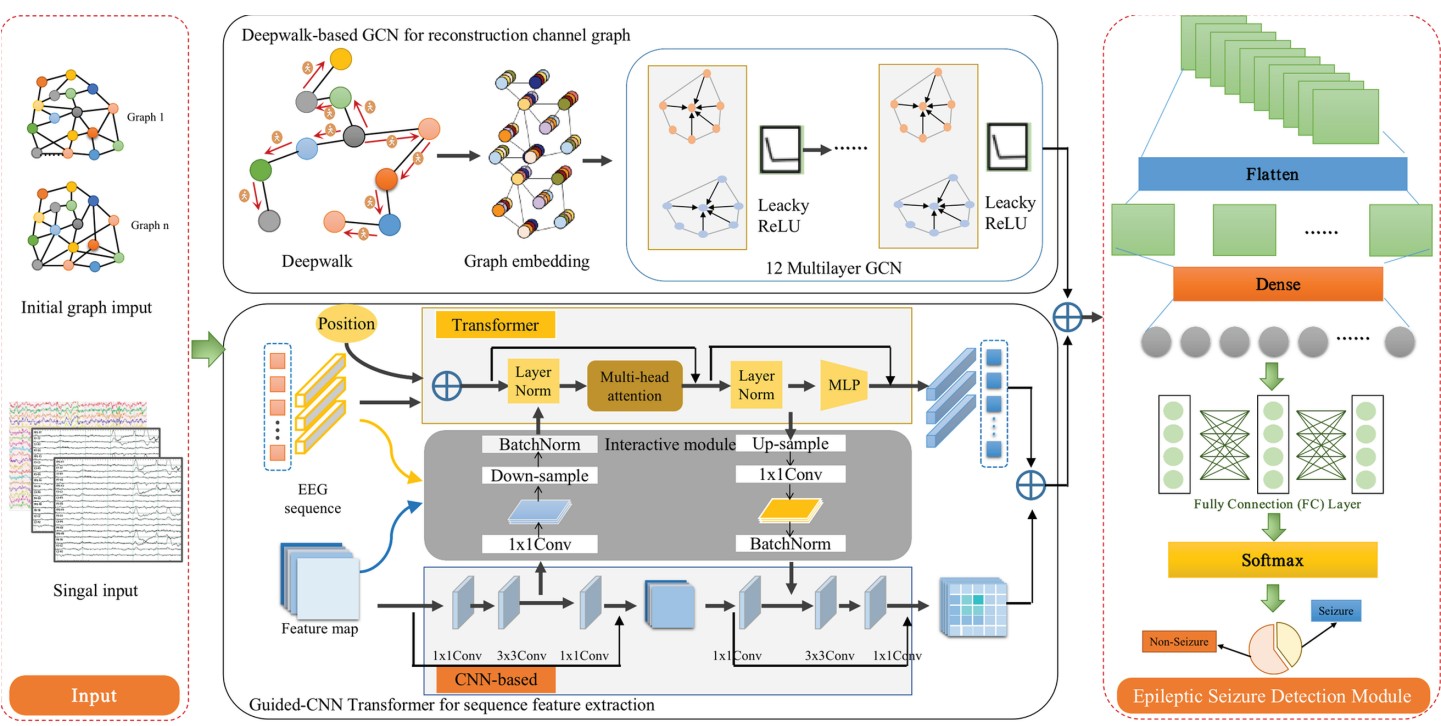

**Fig 1. An overview of proposed Dual-Branch Deepwalk-Transformer Spatiotemporal Fusion Network (Deepwalk-TS) for seizure detection.** The upper shows the spatial branch of Deepwalk-based GCN used to reconstruct the channel graph. The lower shows the temporal branch of Guided-CNN Transformer for sequence feature extraction.

extraction. The prediction module employs FC and Softmax to obtain the prediction results. A detailed description is in the following section.

## Graph structure learning with DeepWalk

Firstly, this branch performs a random walk on the original graph to construct a channel-channel similarity topological space, moving between nodes from one or a series of starting points following specific rules. We give a graph $G = (V, E, X, Y)$ representation. $E$ is the edge of GCN, and $V$ is defined as $\{v_1, v_2, ..., v_{|V|}\}$. $X \in R(N \times T)$ is the feature matrix, $N$ is the number of nodes and $T$ is the dimension. $Y \in R(N \times L)$ is the labeling matrix ($L$ is the dimension). The initial mode is $v_{i_1}$, $W_{v_{i_1}} = \{(v_{i_1}, v_{i_2}), (v_{i_2}, v_{i_3}), ..., (v_{i_{n-1}}, v_{i_n})\}$ is a random walk and a length $n$ rooted at $v_{i_1}$, there is an edge connection between $v_{i_j}$ and $v_{i_{j+1}}$. With a probability p $(0 < p \leq 1)$, the next step moves to a neighboring node. With a probability of $1-p$, it jumps to any node in the graph, with jump probability continuously guiding the random walk for the next step. For each sequence of random walks, it maximizes the co-occurrence probability of vertices in the window $w$,

$$\Pr\left(\{v_{j-w}, ..., v_{j+w}\}/v_j \mid \phi(v_j)\right)$$

$$= \prod_{i=j-w, i \neq j}^{j+w} \Pr\left(v_i \mid \phi(v_j)\right) \tag{1}$$

Fig 2 shows a process of reconstructed graph using random wandering. The order of the walks is marked with a red line and extracted in the next walk graph. After that, we can use

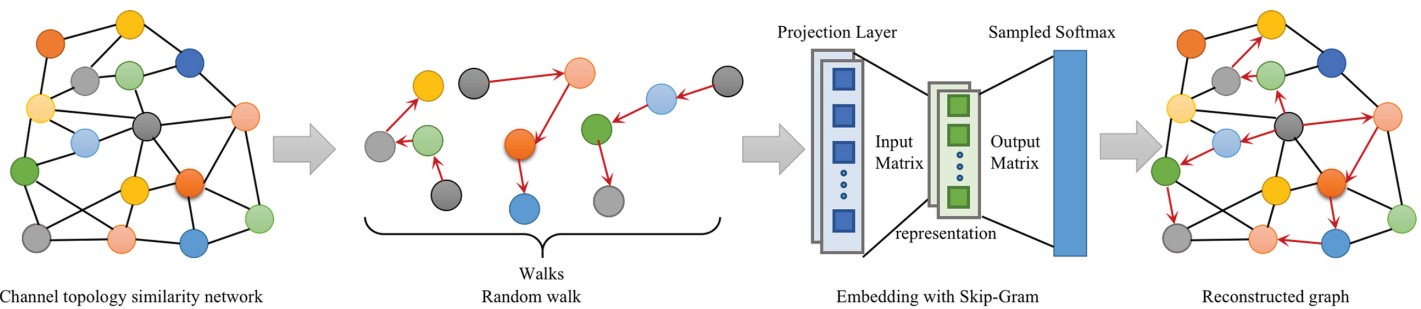

**Fig 2. An illustration of the Deepwalk algorithm operation involves a random walk on the graph, resulting in a "walk" sequence, which is then utilized to construct a graph.**

the obtained "walks" to reconstruct the graph [57]. The image on the right shows a random walk of length 11 nodes (marked in red). The weights $\alpha_k, (k = 1, 2, 3, ..., 11)$ are used to weight the original k-hopping nodes away from the node $c$. The random walk generator takes a random vertex $v_j$ in the graph $G$ as the root of the random walk $w_{vj}$. The topology between brain channels is then reconstructed using the randomness of the Deepwalk. The skipgram algorithm iterates over all possible matches for a random wandering sequence in the window $w$, to obtain the representation matrix $\varphi$. Then, we maximize the probability that this vertex is a neighbor in the walking sequence. But it is infeasible to compute $\Pr\left(v_i \mid \phi\left(v_j\right)\right)$. Therefore, we introduce the Softmax layer to approximate $Pr$, which is expressed as,

$$\Pr\left(v_i \mid \phi\left(v_j\right)\right) = \prod_{k=1}^{[\log|V|]} \frac{1}{1 + e^{-\phi(v_j)\cdot\varphi(b_k)}} \tag{2}$$

where $\varphi\left(b_k\right) \in R_d$ represents the parent node of tree node $b_k$. The path to $v_j$ is identified by the sequence of tree nodes $(b0, b1, ... , \log|v|)$, $b_k = root$ and $\log|v| = v_j$. We utilize the Huffman tree to assign shorter paths to frequently occurring vertices in deepwalk to speed up training time.

Subsequently, we perform graph convolution on the graph reconstructed by DeepWalk to obtain node vectors and graph embeddings. The next step involves using a 12-layer GCN to extract spatial structural features between brain channels. Specifically, the reconstructed graph $G$ is fed into a multi-layer GCN with a parameter sharing strategy to obtain embeddings $\mathbf{Z}_{ct}^{(l)}$ that are shared in different channel spaces. The topology graph $(G_t, X)$ as follows,

$$\mathbf{Z}_{ct}^{(l)} = \mathrm{Re}\,LU\left(\tilde{\mathbf{D}}_t^{-\frac{1}{2}}\tilde{\mathbf{G}}_t\tilde{\mathbf{D}}_t^{-\frac{1}{2}}\mathbf{Z}_{ct}^{(l-1)}\mathbf{W}_c^{(l)}\right), \tag{3}$$

where $\mathbf{W}_c^{(l)}$ is the $l_{th}$ layer weight matrix of GCN and $\mathbf{Z}_{ct}^{(l-1)}$ is the node embedding in the $(l-1)_{th}$ layer and $\mathbf{Z}_{ct}^{(0)} = \mathbf{X}$. $D$ is a diagonal matrix and represents the characteristic dimension of the node. $\widetilde{D}_t$ represents the degree matrix of the vertex. The feature graph $(G_f, X)$ is extract the shared information, the weight matrix $\mathbf{W}_c^{(l)}$ for every layer of GCN as follows,

$$\mathbf{Z}_{cf}^{(l)} = \mathrm{Re}\,LU\left(\tilde{\mathbf{D}}_f^{-\frac{1}{2}}\tilde{\mathbf{G}}_f\tilde{\mathbf{D}}_f^{-\frac{1}{2}}\mathbf{Z}_{cf}^{(l-1)}\mathbf{W}_c^{(l)}\right), \tag{4}$$

where $\mathbf{Z}_{cf}^{(l)}$ is the l-layer output embedding. The shared weight matrix can filter out the shared features from two spaces. Then, the multi-layer GCNs are employed to reorganize the information of the target node and its neighbors.

$$h_v^k = \sigma\left(\sum a_{\tilde{v}u} \cdot h_u^{k-1} \cdot W_k\right) \tag{5}$$

where $h_v^k$ is the hidden representation of node $v$ in the k-th layer. $a_{\tilde{v}u}$ is the numbers on row $v$ and column $u$ of the reconstructed graph matrix $G$, indicating the proximity between nodes $v$ and $u$. $W_k$ is the parameter matrix. $\sigma(-)$ denotes the ReLU function. In short, the branch can combine the graph of the nodes of the harvester at multiple levels to obtain more informative representations. Finally, the obtained representations are fed into the vectors to the downstream classifiers to accomplish the seizure classification task.

### Guided-CNN transformer feature extraction

For the second temporal branch, we introduced a Guided-CNN Transformer method to comprehensively extract both local and global features from EEGs. The branch is composed of three components: a CNN-based module, a Transformer module, and an interactive module, as depicted at the bottom of Fig 1. Firstly, the raw EEGs are input into the CNN feature extractor, which comprises multiple stacked convolutional layers (1×1 Conv, 3×3 Conv, 1×1 Conv) to extract local series features. Secondly, each Transformer block contains a multi-head self-attention mechanism (MHA) and a multi-layer perception block. The dot product between $Q$ and $K$ calculates global location dependencies, which are dotted with $V$ to incorporate these long-range dependencies into the feature. The original self-attention can be calculated,

$$
\begin{aligned}
\text{Att}\left(Q_i, K_i, V_i\right) &= \text{Soft max}\left(\frac{Q_i K_i^T}{d_k}\right) V \\
\text{head}_i &= \text{Attention}\left(Q_i, K_i, V_i\right) \\
MHA(Q, K, V) &= \text{Concat}\left[\text{head}_i\right] W^o
\end{aligned}
\tag{6}
$$

where $i = 1, \ldots, 6$, $Q_I$, $K_I$, and $V_I$ represent the query, key, and value matrix, respectively. Att $(Q_i, K_i, V_i)$ is a weighted. $d_k$ represents $K_i$ dimension. $W^o$ represents the linear transformation of the final output and $head_i$ represents the head of MHA.

Then, we add learnable bias to each self-attention module, which is to enhance the location information,

$$\text{MHA}\left(Q, K', V'\right) = \text{softmax}\left(\frac{QK'^T}{\sqrt{d_k}} + B\right) V' \tag{7}$$

where $K' = Conv(K)$ and $V' = Conv(V)$. Additionally, each head of the MHA module contains a squeezed self-attention layer. The output sequence has the shape of $n \times \frac{d}{h}$ and these $h$ are then concatenated into a $n \times d$ sequence. Then, in the transformer module, the feedforward network (FFN) is a fully connected (FC) layer, expressed as,

$$\text{FFN} = \max\left(0, XW_1 + b_1\right) W_2 + b_2 \tag{8}$$

We have further designed an RFFN to replace the traditional FFN, enhancing both computational efficiency and nonlinear expression capability. The RFFN utilizes a structure similar to the inverted residual block, consisting of two 1×1 conv and a 3×3 depth-wise conv.

$$\text{RFFN} = Conv(DWConv(Conv(X))) + X \tag{9}$$

$$X'_i = \text{LIL}\left(X_{i-1}\right) \tag{10}$$

$$X''_i = \text{MHA}\left(\text{LN}\left(X'_i\right)\right) + X'_i \tag{11}$$

$$X_i = \text{RFFN}\left(\text{LN}\left(X''_i\right)\right) + X''_i \tag{12}$$

where $X'_i$ and $X''_i$ refer to the output of the LIL layer and the SMHSA for block $i$, respectively. LN denotes the layer normalization.

The designed interactive module (inter) integrates the CNN and transformer fusion process. The module includes a transfer feature layer (TF) for keeping invariance, a down/up sample layer for global feature extraction, and an RFFN for enhancing nonlinear expressiveness. The feature $X$ is fed into multi-layer,

$$\text{inter}(X) = \text{Conv}(X) + Up/down(X) \tag{13}$$

We further design a fusion and joint optimization network using dot-multiplication and summation for the proposed spatial-temporal dual-branch method. The final representation vector $z_v$ of node $v$ is obtained and can be sent to the downstream classifier to predict the category vectors $\hat{y}_v$. A Softmax classifier is then employed for classification,

$$\text{Softmax}(z)_i = \frac{\exp\left(z_i\right)}{\sum_{j=1}^{d} \exp\left(z_j\right)} \tag{14}$$

where $\text{Softmax}(z)_i$ is the i-th component of the vector and $d$ is the dimension of $z$.

Finally, guided by labeled data, we optimize the proposed model via back propagation and learn the embedding of nodes for classification. We use cross-entropy as the loss function to train the model,

$$\text{loss} = \sum_v \left(y_v \log \hat{y}_v + \left(1 - y_v\right) \log\left(1 - \hat{y}_v\right)\right) \tag{15}$$

## Experimental

We employ a systematic experimental evaluation, including pre-processing steps, compared with various existing approaches and ablation studies. This analysis dissects the contributions of each branch, validating the effectiveness of their holistic spatiotemporal information integration. To further elucidate the impact of parameters, we present detailed comparative experiments. Additionally, we introduce various visualization techniques to enhance interpretability.

### Datasets and processing

We perform extensive experiments to evaluate the proposed Deepwalk-TS on two widely used EEG datasets: the pediatric patient data from Boston Children's Hospital (CHB-MIT, https://physionet.org/content/chbmit/1.0.0/) and the neurological Siena dataset (Siena). These datasets have distinct origins, enhancing the universality of the model assessment. The CHB-MIT comprises multi-channel EEG recordings from long-term monitoring of pediatric epilepsy patients, featuring various sub-datasets with different time spans and diverse seizure types. The dataset includes 22 subjects (5 males and 17 females) with 23 case records. The Siena primarily records EEGs from 14 patients, including 9 males (aged 25-71) and 5 females

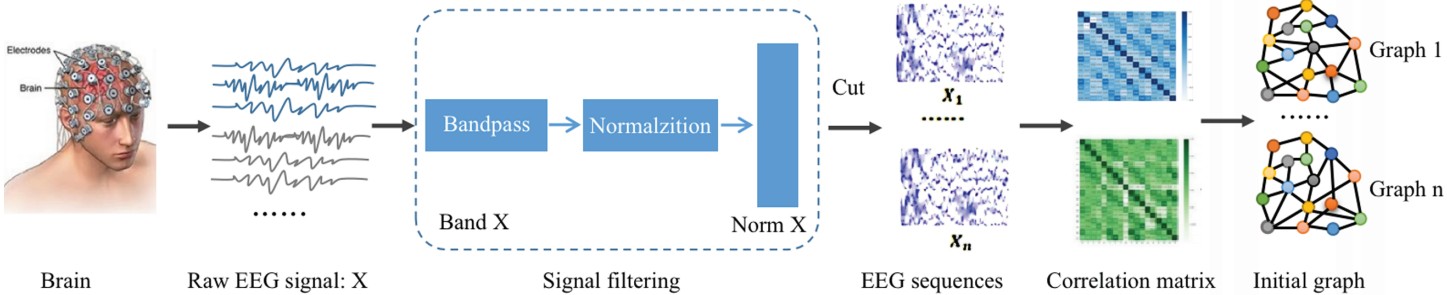

**Fig 3. An illustration of the data preprocessing workflow.**

(aged 20-58). This dataset was recorded at a sampling rate of 512 Hz, and includes multiple channels and diverse patients.

Then, we present the data preprocessing workflow in Fig 3. We applied bandpass filtering to the raw EEGs in the range of 1-40 Hz and standardized the processed multi-channel EEGs (0-1). We employed 1-second or 0.5-second overlapping windows to segment samples of seizure and non-seizure events. For the experiments, we selected the raw EEGs from 16 channels in the CHB-MIT to construct the graph, including channels such as "FP1-F7", "F7-T7", "P7-O1", etc. Subsequently, we combined the EEGs of channels with their corresponding spatial adjacency matrix. We initially formed the graph structure for each case, establishing training and testing datasets.

### Experimental setup and evaluation

To rigorously assess the performance of our proposed Deepwalk-TS model, we conducted a series of comprehensive experiments with carefully structured setups and evaluations. In our experiments, we maintain a training ratio of 1: 9 for positive (epileptic) to negative (non-epileptic) samples. For the patient, we select 400 seconds of positive samples and 3600 seconds of negative samples ($16 \times 256$ dimensionality). We employ a 5-fold cross-validation, and the results are presented as the mean +− standard deviation. The model is trained with a batch size of 128,150 epochs and moment 0.9. For the Deepwalk-based GCN module, we use a two-layer GCN as a graph extractor with identical hidden and output dimensions. where $nhid1 \in \{512, 768\}$ and $nhid2 \in \{32, 128, 256\}$. In our experiments, the tour length $t$ of the random tour is fixed. The traversal sequence randomly selects the neighbors of the last visited node until it reaches the maximum length $t$. $L_2$ norm penalties are applied to the conv kernel parameters in the hierarchical GCN, and $L_1$ is applied to the adjacency matrix. The dropout rate is set to 0.5 during training, while the retention probability is set to 1 during testing. The learning rate varies from 0.0001 to 0.0005, and the coefficients for graph consistency constraints are $\{0.01, 0.001, 0.0001\}$. The experiments are conducted on a 3080Ti GPU with Cuda 10.2, using the PyTorch 1.9.0 framework and updating parameters with the SGD optimizer.

In order to evaluate the performance of our proposed Deepwalk-TS model for epilepsy detection from multiple dimensions, containing Accuracy, Specificity, Sensitivity, Area Under Curve (AUC), ROC curve and F1-score (F1).

$$Accuracy(Acc.) = \frac{TN + TP}{FP + TN + FN + TP} \tag{16}$$

$$Specificity(Spe.) = \frac{TN}{FP + TN} \qquad (17)$$

$$Sensitivity(Sen.) = \frac{TP}{FN + TP} \qquad (18)$$

$$Precision(Pre.) = \frac{TP}{FP + TP} \qquad (19)$$

$$F1 = \frac{2\,Sensitivity \times Precision}{Sensitivity + Precision} \qquad (20)$$

where TP means that both the true label and the predicted label are positive, and FP means that the actual label is negative and the predicted label is positive. TN means that both the true label and the predicted label are negative. FN means that the true label is positive and the predicted label is negative.

## Experimental results

**Single-patient seizure detection results.** We first evaluated the seizure detection results for individual patients using the CHB-MIT dataset, which includes data from all 23 cases. As shown in Table 1, our proposed model dem onstrates outstanding performance, achieving an average accuracy of 99.54%, sensitivity of 99.07%, specificity of 98.87%, F1-Score of 98.84%, and AUC of 99.06%. Notably, accuracy exceeded 98% for 21 patients. However, some variability in performance was observed among patients. Patients such as chb02, chb05, chb08, chb11, chb16, chb18, chb21, chb22 consistently scored perfectly across all evaluation metrics. In contrast, chb12 exhibited slightly lower accuracy at 98.50%, possibly due to less apparent epileptic activity in the selected nine channels. Additionally, Chb14 achieved perfect accuracy and sensitivity but showed lower specificity at 96.83%, likely due to the shorter duration of seizure episodes. Furthermore, notable AUC and F1 were observed for each patient. For instance, chb01 achieved an AUC of 98.99% and an F1-Score of 97.91%, while chb13 demonstrated an even higher AUC of 99.42% and an F1 of 98.86%. The model demonstrated strong stability, with an average specificity of 98.87%, and most patients exhibiting specificity above 98%, indicating its high accuracy in distinguishing non-epileptic states.

We also include this model's additional computational complexity to provide a more comprehensive analysis. Deepwalk-based GCN complexity can be approximated as $O(n \times k \times d)$, where $n$ is the number of nodes ($n = 16$), $k$ is the number of random walks per node ($k = 9$), and $d$ is the dimensionality of the node embeddings. The random walk incurs higher costs with more nodes, walks, and embedding dimensions. Two-layer GCNs typically involve $O(v \times F \times F_1)$, where $v$ is the number of nodes, $F$ and $F_1$ are the original and output feature dimensions. The module's complexity is $O(n \times k \times d) + O(v \times F \times F_1) + O(E \times F_1)$, $E$ is the edges. Through extensive experiments, we have determined that the optimal number of EEG channels is 11, which helps manage the computational overhead. Guided-CNN Transformer complexity is focused on the self-attention mechanisms of Transformer. Q, K, V matrix shape is $n \times d$, $Q \times K_t = (n \times d \times n)$, $scores \times V = softmax(\frac{Q \cdot K^T}{\sqrt{d}}) \times V = O(n^2 d)$, so the complexity is $O(n^2 d + nd) = O(n^2)$, the quadratic complexity comes from the attention calculation between all input token pairs. Then, the two-branch fuse mainly involves element-by-element operations or connections, which does not incur significant computational overhead. The overall complexity is controllable up to $O(n^2)$.

We also conducted single-patient evaluations on the Siena dataset. Table 2 displays the experimental results, showing an average accuracy of 99.18% across all participants. The

**Table 1. Results of CHB-MIT single patient experiments.**

| Case | Acc. | Sen. | Spe. | F1 | AUC |
|------|------|------|------|------|------|
| chb01 | 99.79 | 99.58 | 99.11 | 97.91 | 98.99 |
| chb02 | 100.00 | 100.00 | 100.00 | 100.00 | 100.00 |
| chb03 | 98.61 | 100.00 | 96.42 | 98.18 | 97.32 |
| chb04 | 99.33 | 99.27 | 98.81 | 97.77 | 98.27 |
| chb05 | 100.00 | 100.00 | 100.00 | 100.00 | 100.00 |
| chb06 | 99.52 | 100.00 | 99.58 | 98.04 | 98.12 |
| chb07 | 98.91 | 97.55 | 100.00 | 99.25 | 99.51 |
| chb08 | 100.00 | 100.00 | 100.00 | 100.00 | 100.00 |
| chb09 | 98.81 | 97.24 | 99.29 | 98.23 | 98.93 |
| chb10 | 98.69 | 100.00 | 95.74 | 98.94 | 97.56 |
| chb11 | 100.00 | 100.00 | 100.00 | 100.00 | 100.00 |
| chb12 | 98.50 | 95.64 | 99.39 | 97.82 | 98.69 |
| chb13 | 100.00 | 97.75 | 100.00 | 98.86 | 99.42 |
| chb14 | 100.00 | 100.00 | 96.83 | 95.96 | 96.72 |
| chb15 | 100.00 | 100.00 | 100.00 | 100.00 | 100.00 |
| chb16 | 100.00 | 100.00 | 100.00 | 100.00 | 100.00 |
| chb17 | 98.74 | 100.00 | 95.71 | 96.34 | 99.46 |
| chb18 | 100.00 | 100.00 | 100.00 | 100.00 | 100.00 |
| chb19 | 98.89 | 96.85 | 100.00 | 98.18 | 99.11 |
| chb20 | 99.91 | 99.11 | 100.00 | 99.55 | 99.79 |
| chb21 | 100.00 | 100.00 | 100.00 | 100.00 | 100.00 |
| chb22 | 100.00 | 100.00 | 100.00 | 100.00 | 100.00 |
| chb23 | 99.63 | 95.56 | 93.10 | 98.34 | 96.45 |
| **Mean** | **99.54** | **99.07** | **98.87** | **98.84** | **99.06** |

**Table 2. Results of single-patient experiments on the Sina dataset.**

| Case | Acc. | Sen. | Spe. | F1 | AUC |
|------|------|------|------|------|------|
| PN00 | 100 | 100 | 100 | 100 | 100 |
| PN01 | 99.48 | 99.68 | 99.92 | 99.45 | 98.98 |
| PN03 | 97.89 | 99.77 | 100 | 99.52 | 99.40 |
| PN05 | 100 | 100 | 99.00 | 98.39 | 99.32 |
| PN06 | 98.12 | 97.85 | 98.94 | 97.91 | 97.18 |
| PN07 | 98.60 | 98.65 | 100 | 99.69 | 99.82 |
| PN09 | 97.71 | 98.47 | 98.26 | 98.97 | 97.88 |
| PN10 | 99.20 | 99.50 | 98.75 | 98.62 | 99.01 |
| PN11 | 100 | 100 | 100 | 100 | 100 |
| PN12 | 99.23 | 99.32 | 100 | 98.87 | 99.56 |
| PN13 | 98.56 | 99.20 | 98.40 | 98.98 | 99.12 |
| PN14 | 99.76 | 100 | 97.61 | 97.36 | 98.60 |
| PN16 | 98.93 | 100 | 96.75 | 98.45 | 97.78 |
| PN17 | 99.84 | 99.92 | 99.76 | 99.88 | 99.90 |
| **Mean** | **99.18** | **98.88** | **99.02** | **98.98** | **98.94** |

evaluation results indicate that the majority of cases exhibit a sensitivity above 98%, specificity above 99%, an F1 score with a minimum and average of 98.98%, and AUC values above 98.94%. It is noteworthy that all metrics for PN00, PN11, and PN13 reach 100%. This indicates that the model's predictions are entirely correct, accurately detecting both epileptic seizures and non-epileptic states. Furthermore, the high sampling rate of 512Hz suggests that Deepwalk-TS maintains good performance at different sampling rates.

**Comparison with existing methods.** In Table 3, we conducted extensive experiments to evaluate results compared with recent studies. We categorized methods into four types:

**Table 3. Comparison with existing methods on the CHB-MIT dataset.**

| Types | Author | Method | Accuracy | Sensitivity | Specificity |
|---|---|---|---|---|---|
| Traditional | Gill et al. (2015) [41] | GMM hybrid | 86.93 | 86.26 | 87.58 |
| | Janjarasjitt et al. (2017) [42] | Single Wavelet | 96.87 | 72.99 | 98.13 |
| | Selvakumari et al. (2019) [43] | PCA-Hybrid | 95.63 | 95.70 | 96.55 |
| | Chakraborty et al. (2021) [44] | Multiscale+RF | 95.06 | 98.12 | 99.17 |
| | Li et al. (2021) [45] | WT, EMD+SVM | 97.49 | 97.34 | 97.50 |
| | Chakrabarti et al. (2022) [33] | Random Forster | 91.90 | 94.10 | 89.70 |
| | Li et al. (2023) [46] | SSTFT + FKNN | 98.81 | 98.53 | 99.27 |
| | *Li et al. (2023) [46]* | *Deepwalk + SSTFT* | ***99.01*** | ***98.95*** | ***98.20*** |
| | Amiri et al. (2023) [34] | FSST+LDA | - | 98.44 | 99.19 |
| CNN-based | Yuan et al. (2018) [16] | Spec-CNN | 93.21 | 89.54 | 84.32 |
| | Hossain et al. (2019) [47] | CNN | 98.05 | 90.00 | 91.65 |
| | Fujita et al. (2019) [47] | GAN+1DCNN | 96.15 | 93.53 | 99.05 |
| | Omar et al. (2020) [48] | SOC-CNN | 96.74 | 82.35 | 100.00 |
| | Wang et al. (2021) [49] | S-1D-CNN | 85 | 90.09 | 99.81 |
| | Cimr et al. (2023) [9] | normal-CNN | 96.99 | 97.06 | 96.89 |
| | Xu et al. (2023) [35] | 3D-CNN | - | 95.00 | - |
| | Jiang et al. (2023) [50] | PCA-SVM | 96.67 | 97.72 | 95.62 |
| | Xiao et al. (2024) [36] | SLAM | 97.07 | 96.68 | 97.42 |
| | *Zhao et al. (2023) [37]* | *CNN+Transf* | ***98.76*** | ***97.70*** | ***97.60*** |
| LSTM-based | Shahbazi et al. (2018) [51] | CNN + LSTM | 95.51 | 95.14 | 94.86 |
| | Huang et al. (2019) [52] | CNN-BiRNA | - | 92.88 | 93.94 |
| | Hu et al. (2020) [18] | BiLSTM | - | 93.61 | 91.85 |
| | Yao et al. (2021) [38] | LSTM | 88.63 | 87.00 | 88.63 |
| | *Asma et al. (2023) [39]* | *LSTM+att* | ***96.48*** | ***96.28*** | ***96.88*** |
| GCN-based | Chen et al. (2020) [53] | EGCN | 98.35 | - | - |
| | He et al. (2020) [54] | GCN+LSTM | 98.52 | 97.75 | 94.34 |
| | Zhao et al. (2021) [13] | A-GRN+fl | 98.70 | 99.16 | 98.66 |
| | Jibon et al. (2023) [56] | LGCN-Den | 98.00 | 97.84 | 98.33 |
| | Zheng et al. (2021) [55] | HGCN | 99.40 | 99.53 | 88.87 |
| | *Zheng et al. (2021) [55]* | *HGCN + TS* | ***99.44*** | ***98.51*** | ***92.36*** |
| | **Ours** | **Deepwalk-TS** | **99.54** | **99.07** | **98.87** |

machine learning, CNN-based, LSTM-based, and GCN-based. We first compared Deepwalk-TS with several traditional machine learning approaches, outperforming GMM, Wavelet, PCA-Hybrid, and Random Forest across all metrics, showing a remarkable 12.61% improvement in accuracy over GMM's 86.93%. Then, we compared Deepwalk-TS with CNN-based methods. The CNN+Transformer achieved 98.76% accuracy, 97.70% sensitivity, and 97.60% specificity, respectively, highlighting the effectiveness of our method in capturing intricate patterns in EEG data. Given the sequential feature of EEGs, LSTM-based detection methods have been widely studied. We further compare our approach with classical methods such as CNN+LSTM and LSTM+attention. While LSTM achieved an accuracy of 88.63%, Deepwalk-TS surpassed it with an impressive 10.91% improvement. This improvement is because LSTM primarily focuses on single sequences, making it challenging to integrate information directly. In contrast, Deepwalk-TS integrates the graph structure between channels established by Deepwalk theory. We also compared Deepwalk-TS with GCN-based methods, which demonstrates significant improvements. For instance, against A-GRN+focal loss with an accuracy of 98.70%, Deepwalk-TS achieves an improvement of 0.84%. This indicates that our multi-channel modeling effectively captures spatial relationships.

In experiments exploring the effectiveness of existing feature extraction methods for our module, we combined the optimal SSTFT features with our Deepwalk, achieving an accuracy of 99.01%, sensitivity of 98.95%, and specificity of 98.20%. This demonstrates that both our graph-based feature extraction and existing features contribute to improved epilepsy detection. Similarly, when we combined the representative HGCN method with our TS, we achieved an accuracy of 99.44%, marking a significant improvement over the original method. This proves that incorporating the described seizure-related features into their models enhances their performance.

To further validate the performance, we compared the AUC of our method with the top four performing methods. To present epilepsy detection results intuitively, we further used the ROC graph visualization for assessment. The Fig 4 depicted the average ROC curve of the Deepwalk-TS, with an average AUC of 98.87% across 23 patients, illustrating the strong capability in real-world detection scenarios. In clinical epilepsy monitoring, Accuracy more than 95% is sufficient to alert doctors or instruments to make timely diagnostic interventions. Our proposed Deepwalk-TS model has achieved a very satisfactory 99.54% Accuracy in the test phase. While 100% accuracy is rare in real-world tasks like seizure detection due to noise and variability in the data. The proposed Deepwalk-TS model's sensitivity of 99.07% ensures timely seizure detection, while specificity of 98.97% effectively distinguishes seizure from non-seizure events, minimizing false positives.

**Qualitative analysis.** First, we delve into graph structure analysis and visualization. We reveal the graph structure changes of the spatial graph reconstruction branch and the stochastic generation capability of the Deepwalk-TS model. To showcase the significantly diverse graph structures generated by different patients, we randomly selected and displayed structures from two patients (ch00, ch05) with distinct random walks in Fig 5. Calculating similarity by averaging Manhattan distances demonstrates the ability to generate dynamic graphs at

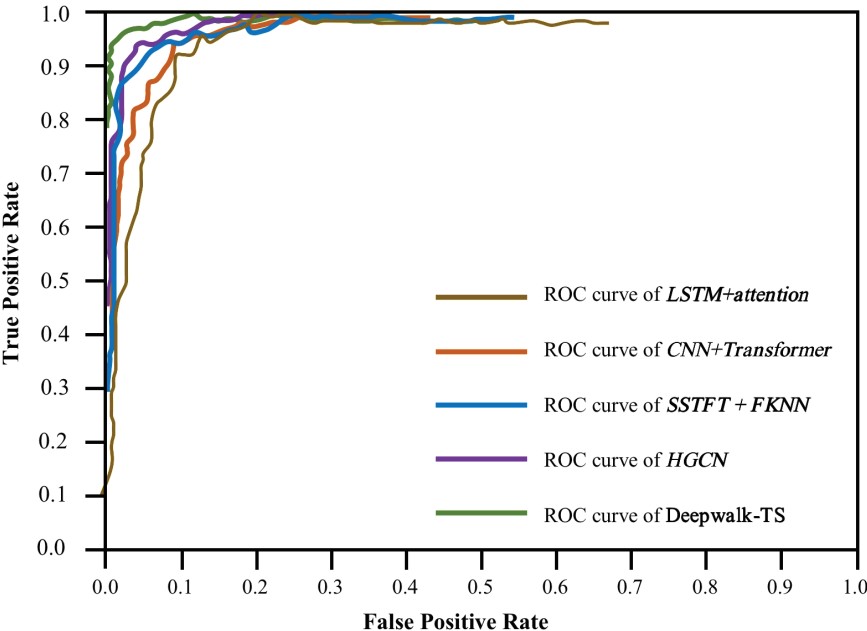

**Fig 4. ROC curve comparing our method with advanced performance methods.**

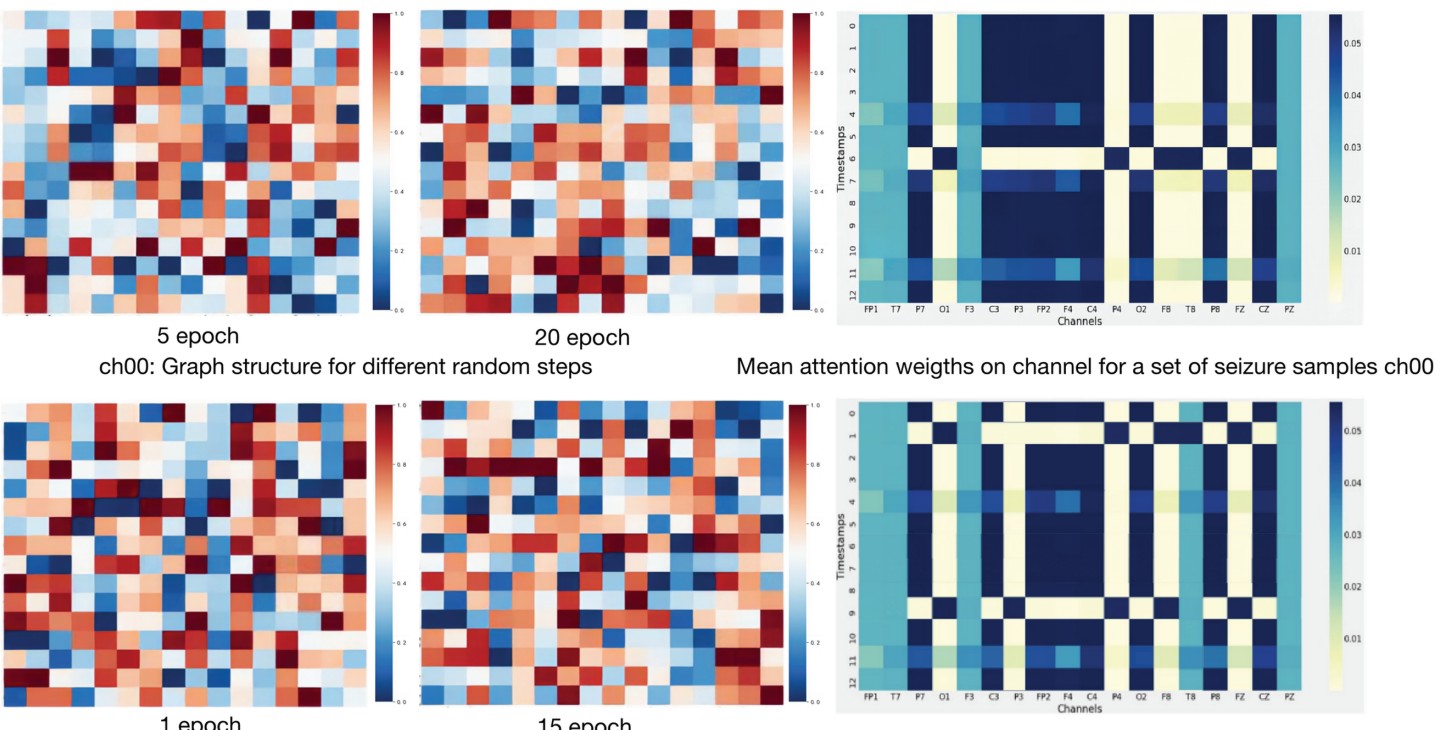

**Fig 5. Example of the process of constructing a map of multichannel EEGs.**

different points in time randomly. The last column represents the final weight matrix of channel attention. There is a noticeable difference in the relationships between channels for each sample.

Additionally, we visualized the epilepsy test results using t-SNE technology to highlight the feature embedding capability of the Deepwalk-TS model, as shown in Fig 6. t-SNE is a non-linear dimensionality reduction method that effectively reduces high-dimensional feature data to a two-dimensional space. We displayed the raw EEG signals and the embedded data output from the model's fully connected (FC) layer for all samples. Different colors represent normal (blue) and epileptic seizure (orange) signals.

In Fig 6(a), the original feature space shows little distinction between normal and epileptic brain signals. However, Fig 6(b) presents the t-SNE plot of the data embeddings obtained after training Deepwalk-TS, derived from the model's FC layer output. This visualization clearly separates the two classes in the two-dimensional space, improving the interpretability of the model's performance in detecting epileptic seizures.

Finally, to address the reviewer's suggestion, we have added the learning curves, specifically the accuracy and loss curves, of the DeepWalk-TS model in the experimental section. These curves provide a clear visualization of the training dynamics, model convergence, and overall performance during the training and validation phases. As shown in Fig 7, the model exhibits stable convergence, as evidenced by the consistent decrease in the loss function over time, indicating effective optimization. The smooth decline in both training and validation

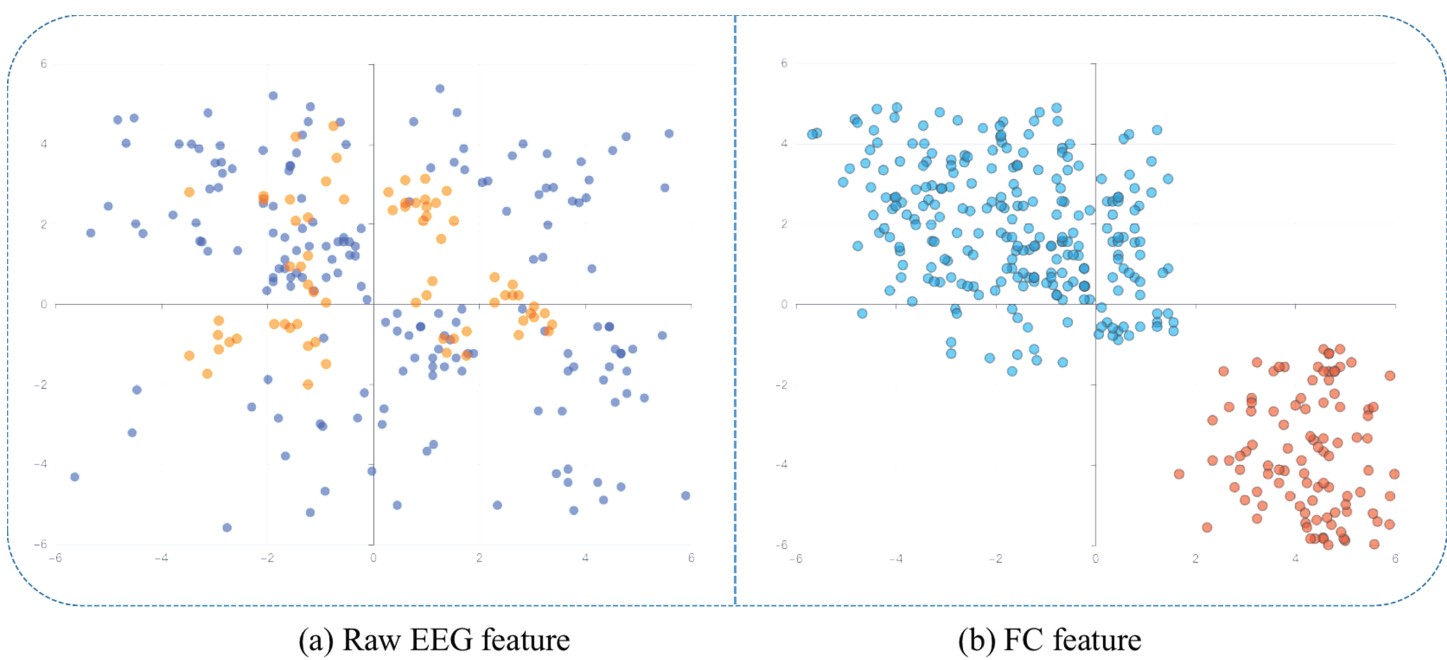

(a) Raw EEG feature  (b) FC feature

**Fig 6. t-SNE visualization of the original and model-embedded distributions of the test data.** Different blue and orange represent the characteristic sample points of normal and epileptic seizures, respectively. (a) Original data distribution. (b) Distribution of feature embeddings for model prediction results.

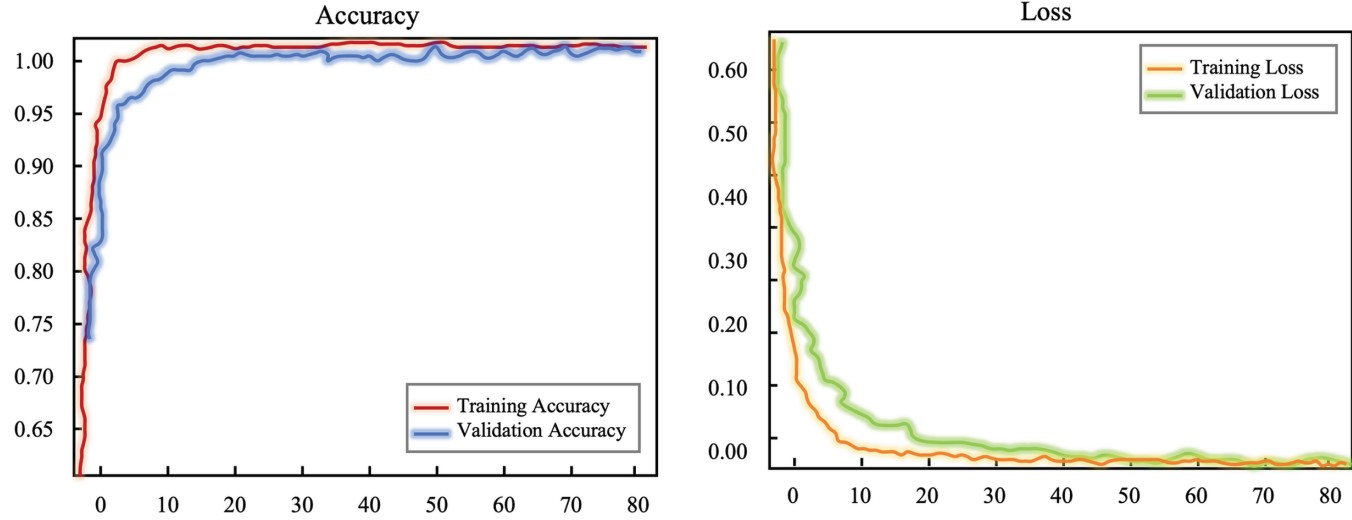

**Fig 7. Accuracy and loss curves for DeepWalk-TS model during the training and validation process.**

loss suggests efficient learning. Typically, a large gap between training and validation loss signals overfitting, while a minimal gap indicates better generalization. In our results, the validation loss closely follows the training loss, confirming that overfitting is minimal and does not compromise the model's overall performance.

## Ablation studies

In this section, we conducted ablation studies on Deepwalk-TS to explore the relationships between spatial channels. We systematically analyzed the impact of the spatial and temporal branches and the influence of each component on overall performance. Firstly, we performed experiments by removing the Guided-CNN transformer, indicating a decline in performance and highlighting its significant positive impact on model performance. Secondly, we conducted experiments by removing the graph embedding Deepwalk-based GCN. The results showed a weakened ability of the model to distinguish between normal and epileptic signals upon removing graph embedding. The specific results of the ablation experiments are detailed in the table below, outlining performance metrics under different model configurations.

As shown in Fig 8, Deepwalk-TS demonstrates the best performance across all evaluation metrics. As a baseline model, GCN achieved an accuracy of 97.89%, sensitivity of 97.32%, specificity of 98.56%, F1-Score of 97.98%, and AUC of 98.10%. However, GCN slightly lags behind other models in terms of accuracy and AUC, suggesting further room for improvement in exploring the spatial relationships of EEG channels. The designed Deepwalk submodule shows a significant improvement over GCN with 98.76% accuracy, 98.42% sensitivity and 99.12% specificity. It shows the effectiveness of the node embedding in learning channel relationships.

Furthermore, the Transformer temporal branches achieve an accuracy of 98.45%, a significant improvement over traditional LSTM. Its self-attention mechanisms provide a significant

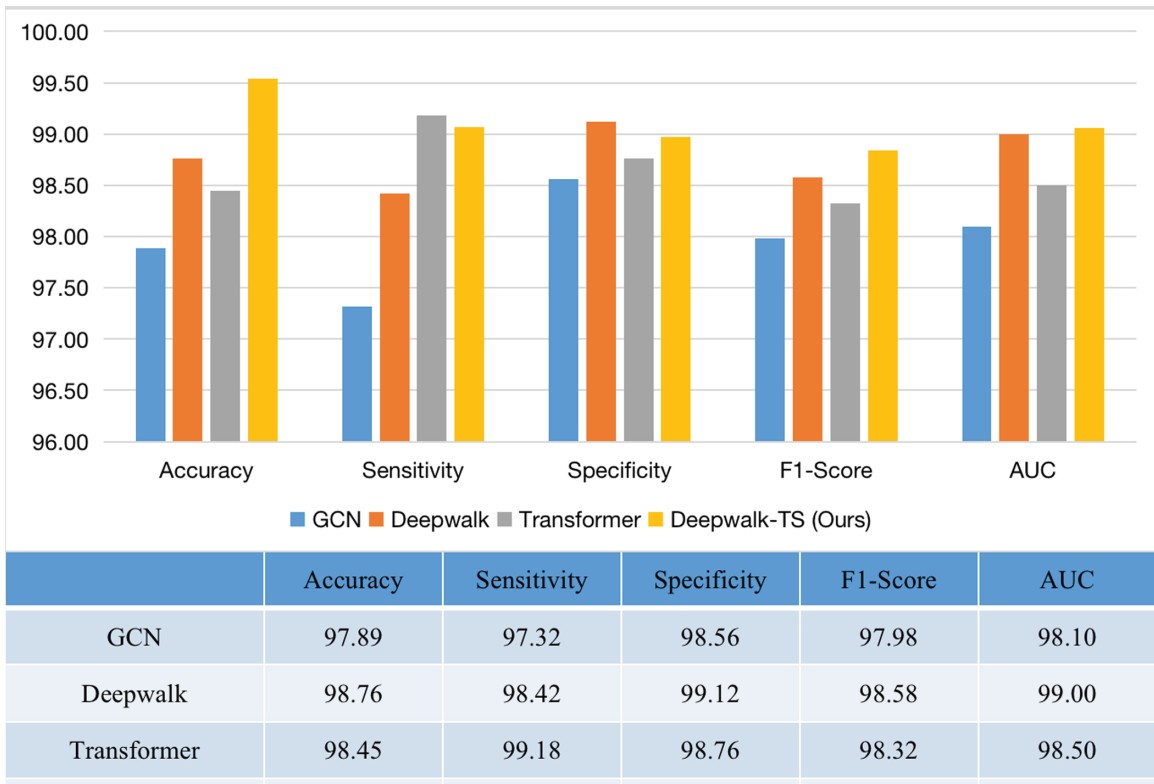

|  | Accuracy | Sensitivity | Specificity | F1-Score | AUC |
|---|---|---|---|---|---|
| GCN | 97.89 | 97.32 | 98.56 | 97.98 | 98.10 |
| Deepwalk | 98.76 | 98.42 | 99.12 | 98.58 | 99.00 |
| Transformer | 98.45 | 99.18 | 98.76 | 98.32 | 98.50 |
| **Deepwalk-TS (Ours)** | **99.54** | **99.07** | **98.97** | **98.84** | **99.06** |

**Fig 8. Visual histogram of ablation test performance.**

advantage, delivering outstanding sensitivity of 99.18% and specificity of 98.76%. Compared to Deepwalk, our Deepwalk-TS demonstrates an improvement of 0.78% in accuracy, 0.26% in F1-Score, and 0.86% in AUC. The Transformer module positively influences overall performance, particularly in enhancing temporal information capture and significantly improving AUC. The model components work synergistically, demonstrating a well-designed and effective overall architecture. Their combined strengths enable Deepwalk-TS to excel in epilepsy detection, with Deepwalk enhancing spatial relationships between EEG channels and the Transformer improving temporal information capture.

## Discussion

Our study on epileptic seizure detection using the Deepwalk-TS method has yielded several notable findings. This section provides an in-depth discussion of the primary results, real-world performance, strengths and limitations, and potential directions for future research.

Our results demonstrate that the Deepwalk-TS model significantly outperforms both traditional and contemporary models in epileptic seizures detection across multiple datasets. The dual-branch architecture, integrating the Deepwalk-based GCN for spatial analysis and the Guided-CNN Transformer for temporal feature extraction, has shown to be highly effective. Compared to machine learning approaches like GMM and PCA-Hybrid models, Deepwalk-TS shows remarkable improvements in accuracy, sensitivity, and specificity. Unlike those methods [33,34], Deepwalk-TS leverages advanced deep learning techniques to automatically extract meaningful features. CNN-based methods, while effective in capturing local features from EEG signals, often struggle to capture long-term dependencies and spatial relationships [35–37]. Similarly, LSTM-based methods, which are adept at handling sequential data, often neglect the spatial structure of EEG signals [38,39].

The Deepwalk-TS has several notable strengths. First, its dual-branch architecture effectively integrates spatial and temporal features, addressing key challenges in EEGs analysis. The Deepwalk-based spatial branch dynamically captures high-order relationships among EEG channels through adaptive graph construction. Second, the Guided-CNN Transformer temporal branch achieves spatial feature extraction and global attention dependencies effectively. This combination enables the model to extract meaningful sequential patterns, significantly enhancing its sensitivity to seizure-related signals. Third, extensive experiments demonstrate that Deepwalk-TS achieves performance on multiple datasets, underscoring its generalizability and effectiveness across diverse patient populations and data sources. Unlike existing methods that focus on either spatial or temporal dimensions, our dual-branch architecture integrates both, enabling the model to capture complex EEG channel relationships and temporal dependencies.

Despite these promising outcomes, there are several limitations to consider. It is still affected by the inherent variability in individual physiological characteristics across different patients, which impacts the generalizability of the model to unseen data. Furthermore, the complexity of EEG sequences and the high dimensionality of the substantial labeled data pose challenges to computational efficiency and scalability. While the visual interpretability of graph structures and channel attention is a strength, the framework lacks detailed mechanisms for clinical feedback loops. Finally, while our method performs well in offline scenarios, real-time detection may present computational challenges.

The Deepwalk-TS method shows great application potential in online epilepsy detection and real-world deployment. For real-time detection, the spatiotemporal information integration enables the algorithm to process continuous EEGs effectively, achieving highly accurate

seizure detection. The framework can be optimized for low-latency detection through techniques such as model pruning, lightweight adaptations, and parallel computations. Regarding integration into closed-loop systems, further optimization is required to enhance real-time performance and feedback integration. The method promises seamless integration into clinical workflows, enabling early epilepsy intervention and real-time feedback in closed-loop systems.

Future research should focus on several key areas to further enhance the capabilities and applicability of the Deepwalk-TS model. Firstly, refining the spatial exploration capabilities can lead to even better utilization of the inherent relationships between EEG channels. Enhancing the robustness of the model to handle inter-individual variability via multi-view representation learning. To further enhance the model's performance across diverse datasets, we will involve exploring advanced augmentation techniques, such as transfer learning, and domain adaptation. Additionally, techniques such as model pruning and unified representation will be employed to reduce computational complexity while maintaining interpretability in clinical settings. Adapting real-time EEG processing for emotion detection offers significant potential in healthcare, human-computer interaction, and education. Extending our work to emotion classification could lead to systems that dynamically respond to emotional states, improving decision-making in real-world scenarios [40]. Focusing on these areas will not only enhance the performance of the model in epileptic seizure detection but also provide a foundation for extending its application to other neurological disorders.

## Conclusion

Our investigation into epileptic seizure detection underscores the importance of addressing persistent challenges, such as underutilized spatial relationships among EEG channels, inter-individual physiological variability, and complexities in EEG sequences. To improve the accuracy of epilepsy detection, we introduced the innovative Dual-Branch Deepwalk-Transformer Spatiotemporal Fusion Network (Deepwalk-TS), specifically designed to address these issues in EEGs. This model integrates a Deepwalk-based GCN for spatial analysis and a Guided-CNN Transformer for temporal sequence feature extraction. Our experiments on multiple datasets show that Deepwalk-TS outperforms existing models and significantly improves the accuracy of epilepsy detection. Ablation experiments affirm the rational and effective design of our model, demonstrating the complementary nature of its various components. In summary, our proposed Deepwalk-TS model, with its dual-branch architecture and fusion of spatiotemporal features, represents a promising advancement in epileptic seizure detection. Future research should refine spatial exploration, enhance model robustness to inter-individual variability, and optimize performance across datasets. Reducing computational complexity with pruning will improve the model's applicability and interpretability, enabling broader use in epilepsy detection and other neurological disorders.

## Acknowledgments

We gratefully acknowledge the valuable discussions from our colleagues at Nanchang Institute of Technology, whose expertise contributed to developing seizure detection. We also thank the relevant research teams for providing public datasets and the college for offering experimental facilities. Our appreciation extends to the anonymous reviewers for their constructive feedback on earlier versions of this paper.

## Author contributions

**Conceptualization:** Xiaobing Deng.

**Funding acquisition:** Xiaobing Deng.

**Investigation:** Xiaobing Deng.

**Methodology:** Xiaobing Deng.

**Software:** Xiaobing Deng.

**Visualization:** Xiaobing Deng.

**Writing – original draft:** Xiaobing Deng.

**Writing – review & editing:** Xiaobing Deng.

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
