## [Decision Letter · Decision Letter 0]

3 Jan 2025

PONE-D-24-53519A Novel Dual-Branch Network for Comprehensive Spatiotemporal Information Integration for EEG-based Epileptic Seizure DetectionPLOS ONE

Dear Dr. Deng,

Thank you for submitting your manuscript to PLOS ONE. After careful consideration, we feel that it has merit but does not fully meet PLOS ONE’s publication criteria as it currently stands. Therefore, we invite you to submit a revised version of the manuscript that addresses the points raised during the review process.

We look forward to receiving your revised manuscript.

Kind regards,

Yuvaraj Rajamanickam, Ph.D

Academic Editor

PLOS ONE

Journal Requirements:

“This study was supported by the project foundation design and implementation of intelligent analysis and management platform for the urban economy (GJJ2202706), Xiaobing Deng.”

“This study was supported by the project foundation design and implementation of intelligent analysis and management platform for urban economy (GJJ2202706).”

“This study was supported by the project foundation design and implementation of intelligent analysis and management platform for the urban economy (GJJ2202706), Xiaobing Deng.”

Additional Editor Comments (if provided):

Comments:

Please discuss whether the application of the algorithm is possible for online detection (i.e., close loop system).

What are the strengths and weaknesses of the proposed framework? Please include them in the paper.

What about the computational complexity of the proposed method? Some description related to computational complexity is required in the paper.

There is a huge amount of literature regarding EEG-derived fingerprints of seizure detection. Do authors consider that the incorporation of already described seizure-related features into their model could improve their results? Some discussion on that is needed. Without this more rigorous comparison with published work, this work is just another incremental addition.

The authors never state what level of performance would be considered adequate for a “real-world scenario” system that is useful and valid.

How the proposed emotion classification system be useful in the real-world scenario? Need to discuss.

Reviewers' comments:

Reviewer's Responses to Questions

**Comments to the Author**

1. Is the manuscript technically sound, and do the data support the conclusions?

Reviewer #1: Yes

Reviewer #2: Yes

Reviewer #3: Yes

2. Has the statistical analysis been performed appropriately and rigorously? 

Reviewer #1: Yes

Reviewer #2: Yes

Reviewer #3: Yes

3. Have the authors made all data underlying the findings in their manuscript fully available?

Reviewer #1: Yes

Reviewer #2: Yes

Reviewer #3: Yes

4. Is the manuscript presented in an intelligible fashion and written in standard English?

Reviewer #1: Yes

Reviewer #2: Yes

Reviewer #3: Yes

5. Review Comments to the Author

Reviewer #1: The authors have performed a study on a deep-learning based model to detect epileptic seizures. The paper is well-written and the proposed methodology is robust.

Some minor points need to be revised.

Introduction:

- page 1: the authors state that “early diagnosis of seizures is important for patients with refractory epilepsy to prevent irreversible damage to the patient’s brain”. This sentence makes too many assumptions in implying that single epileptic seizures can damage the brain irreversibly. Seizure detection is an important goal, but this sentence needs to be reformulated.

Results:

- Page 7: section 4.1 and 4.2 should be moved to Method section, since they do not represent a result of the current research.

Discussion:

- Page 14: the authors have analyzed data from two different databases, one of them being represented by a pediatric population. This may influence the results and should be discussed.

Reviewer #2: 1. Please define SOTA in the abstract.

2. Please mention the details in the abstract.

3. Please add paper organization at the end of the introduction.

4. The related work section is missing.

5. Fig.6, please indicate the blue and orange colours.

6. Limitations of the proposed system should be discussed in the paper.

7. The learning curve (Accuracy and loss) of the DL models is missing from the paper.

Reviewer #3: Comment to author(s):

I have gone through this article. It is based on epileptic seizure detection by employing a DeepWalk algorithm on EEG datasets. The overall study has great potential to add significant value to relevant fields. However, some points should be considered before the final acceptance of this article.

1. The evaluation scores were significantly high with an accuracy of 99.54%. Why does it not reach 100%, and how can overfitting be understood in the proposed work? It is possible to add a description related to these concerns to the Discussion section, or wherever possible in the manuscript, it is feasible.

2. Please mention the acronyms in full form when they are first used in the manuscript, such as "GMM," on page 9.

3. Please check Page no. 5, equation 2 and equation 3, the variables "bk" and "Dt" are defined somewhere or not. Also, check all the typos if they are available in the manuscript, such as in page no. 8, section 4.3.1, "...proposed model dem onstrated..."

Overall, the manuscript has great potential and the proposed work is highly valuable. We look forward to getting this article to the next stage of article processing and acceptance.

6. PLOS authors have the option to publish the peer review history of their article (what does this mean?). If published, this will include your full peer review and any attached files.

Reviewer #1: **Yes: **Umberto Aguglia

Reviewer #2: No

Reviewer #3: No

---

## [Author Response · Author response to Decision Letter 1]

3 Feb 2025

Response to Reviewers

Dear Editor and Reviewers:

System Response to Reviewers lacks formula and image modification, for details, please refer to the uploaded file entitled "Response to Reviewers". Thank you for your letter and the reviewers’ comments concerning our manuscript entitled “A Novel Dual-Branch Network for Comprehensive Spatiotemporal Information Integration for EEG-based Epileptic Seizure Detection (No. PONE-D-24-53519)”. We deeply appreciate the constructive comments, helpful critiques, and favorable assessments from all reviewers. Those comments are valuable and have significantly contributed to the revision and improvement of our paper. We have carefully reviewed the comments and made the necessary modifications to the resubmitted manuscript. Below are our detailed point-by-point responses to each comment.

Editor’s Comments:

Thank you for your thorough review of our manuscript, bring your constructive feedback and positive comments on the proposed Deepwalk-TS algorithm for epilepsy detection. Below, we address the issues you raised and outline the changes made to improve the paper’s quality and suitability for publication.

1.Please discuss whether the application of the algorithm is possible for online detection (i.e., close loop system).

We have discussed the online application of Dual-Branch Deepwalk Transformer Spatiotemporal Fusion Network (Deepwalk-TS) in detail in the "Discussion" section. It can be determined that method shows great application potential in online epilepsy detection. The algorithm innovatively integrates the spatial information analysis of Deepwalk-based and the temporal feature extraction of Guided-CNN Transformer, dynamically and efficiently processes EEGs, and has achieved accurate and timely epilepsy detection. Although the real-time performance and feedback integration need to be further optimized, the proposed method provides a solid foundation for advanced clinical systems to support online epilepsy detection in a timely manner.

For real-time detection, the spatiotemporal information integration enables the algorithm to process continuous EEGs effectively, achieving highly accurate seizure detection. The Deepwalk-based graph construction dynamically captures spatial relationships across EEG channels, while the Transformer module efficiently processes time-series data, ensuring real-time detection capability. Although current implementations may require minor delays for feature extraction, the framework can be optimized for low-latency detection through techniques such as model pruning, lightweight architecture adaptations, and parallel computations. The model is well-suited for online streaming data, making it feasible for real-time monitoring.

Regarding integration into closed-loop systems, our study primarily focuses on algorithmic design and localized detection using multiple datasets. The method's robust ability to analyze spatiotemporal EEG features ensures reliable seizure detection, even in complex clinical scenarios. While the method emphasizes accuracy and strong performance in static evaluations, further optimization is required to enhance real-time performance and feedback integration. As part of a feedback loop, the system can process real-time EEG data streams to continuously update the prediction model, leveraging adaptive learning techniques such as incremental training or online fine-tuning. Additionally, incorporating active learning or adaptive model updates based on real-time feedback could further improve the model's ability to adapt to new seizure patterns. The proposed framework promises seamless integration into clinical workflows, enabling early intervention and real-time feedback in closed-loop systems for epilepsy management.

2.What are the strengths and weaknesses of the proposed framework? Please include them in the paper.

We have added a detailed discussion of the strengths and weaknesses of our method in the revised manuscript to enhance the clarity and depth of the paper.

The Deepwalk-TS has several notable strengths: First, its dual-branch architecture effectively integrates spatial and temporal features, addressing key challenges in EEGs analysis. The Deepwalk-based spatial branch dynamically captures high-order relationships among EEG channels through adaptive graph construction, ensuring that spatial dependencies are preserved and utilized efficiently. Second, the Guided-CNN Transformer temporal branch leverages CNN’s spatial feature extraction capabilities and the Transformer’s global attention mechanism to model temporal dependencies effectively. This combination enables the model to extract meaningful sequential patterns, significantly enhancing its sensitivity to seizure-related signals. Third, extensive experiments demonstrate that Deepwalk-TS achieves state-of-the-art performance on multiple datasets, underscoring its generalizability and effectiveness across diverse patient populations and data sources. Additionally, the model’s design facilitates interpretability by allowing visual analysis of learned graph structures and channel attention mechanisms, providing valuable insights for clinical applications.

While Deepwalk-TS exhibits high accuracy and robust feature extraction capabilities, there are several limitations to consider. It is still affected by the inherent variability in individual physiological characteristics across different patients, which impacts the generalizability of the model to unseen data. Another limitation is the reliance on substantial labeled data, as the method's performance with raw, unsegmented signals is untested. Furthermore, while the visual interpretability of graph structures and channel attention is a strength, the framework lacks detailed mechanisms for clinical feedback loops.

Future research should focus on several key areas: Enhancing the robustness of the model to handle inter-individual variability via multi-view representation learning. To further enhance the model's performance across diverse datasets, we will explore advanced techniques such as data augmentation, transfer learning, and domain adaptation. Additionally, techniques such as model pruning and unified representation will be employed to reduce computational complexity while preserving accuracy and interpretability. Focusing on these areas will not only enhance the performance and applicability of the method in epileptic seizure detection but also provide a foundation for researchers to extend its use to other neurological disorders and related fields. We have incorporated these discussions in the revised manuscript to provide a more balanced and comprehensive overview of the framework’s strengths, weaknesses, and directions for future improvements.

3.What about the computational complexity of the proposed method? Some description related to computational complexity is required in the paper.

Thank you for raising the important point regarding the computational complexity of our proposed method. We included additional calculations in Experimental Results subsection to provide a more comprehensive analysis. The computational complexity of this model primarily stems from two key components: the Deepwalk-based GCN and the Guided-CNN Transformer.

Deepwalk-based GCN complexity can be approximated as O(n×k×d), where n is the number of nodes (n = 16), k is the number of random walks per node (k = 9), and d is the dimensionality of the node embeddings. The random walk incurs higher costs with more nodes, walks, and embedding dimensions. Two-layer GCNs typically involve O(v×F×F1), where v is the number of nodes, F and F1 are the original and output feature dimensions. The module's complexity is O(n×k×d)+O(v×F×F1)+ O(E×F1), E is the edges. Through extensive experiments, we have determined that the optimal number of EEG channels is 11, which helps manage the computational overhead. Guided-CNN Transformer complexity is focused on the self-attention mechanisms of Transformer. Q, K, V matrix shape is n×d, Q×Kt = (n×d×n), scores×V = softmax(Q×Kt /d)×V = O(n2d), so the complexity is O(n2d + nd) = O(n2), the quadratic complexity comes from the attention calculation between all input token pairs. Then, the outputs of the GCN space and Transformer time branches are fused, which mainly involves element-by-element operations or connections, which does not incur significant computational overhead. During training, the two branches are jointly optimized through techniques such as balanced loss function and mini-batch gradient descent, and the overall complexity is controllable up to O(n2). We have expanded the discussion of computational complexity in the revised paper, with a clearer breakdown of each component.

4.There is a huge amount of literature regarding EEG-derived fingerprints of seizure detection. Do authors consider that the incorporation of already described seizure-related features into their model could improve their results? Some discussion on that is needed. Without this more rigorous comparison with published work, this work is just another incremental addition.

Our work focuses on developing an end-to-end framework, the Deepwalk-TS, which automatically learns spatiotemporal features directly from raw EEG data. This approach eliminates the dependency on handcrafted features, allowing the model to autonomously capture high-order spatial relationships and temporal dependencies. Below, we address the existing seizure features and works, results comparisons with existing research, and the incremental contributions of our proposed model.

In the revised manuscript, we have added a Related Works section, providing an overview of feature extraction and EEG-based seizure detection methods. In this paper, we focused on a deep learning-based approach that learns both spatial-temporal relationships between EEG channels, which is an area that has not been fully explored in the current works. Then, we revised the manuscript to clarify and expand on the integration of existing seizure-related features in the Experiment section. We conducted additional experiments incorporating mainstream handcrafted-based and GCN-based epilepsy detection methods, fusing features with representative methods like SSTFT and HGCN in Table 3. The results of these experiments have been updated in Table 3 of the revised manuscript.

We combined the optimal SSTFT features with our Deepwalk, achieving an accuracy of 99.01%, sensitivity of 98.95%, and specificity of 98.20%. This demonstrates that both our graph-based feature extraction and existing features contribute to improved epilepsy detection. Similarly, when we combined the representative HGCN method with our TS, we achieved an accuracy of 99.44%, marking a significant improvement over the original method. This proves that incorporating the described seizure-related features into their models enhances their performance. We acknowledge that integrating existing features could enhance the model's generalizability and robustness, providing complementary value and improving its practical efficacy.

While deep learning-based seizure detection is a growing research field, the novelty of Deepwalk-TS lies in its holistic integration of spatiotemporal features. Unlike existing methods that primarily focus on either spatial or temporal dimensions, our dual-branch architecture combines both, allowing the model to capture intricate relationships across EEG channels while effectively modeling temporal dependencies. This unified approach enhances the model's adaptability to complex and heterogeneous datasets, improving its robustness and accuracy in seizure detection. Therefore, our work should not be viewed as merely an incremental addition to existing research, but rather as a significant exploration in epileptic seizure detection.

5.The authors never state what level of performance would be considered adequate for a “real-world scenario” system that is useful and valid.

The reviewer's comment has inspired us to recognize the importance of defining performance thresholds for real-world epilepsy detection, which are crucial for assessing the practical applicability of our method. A system should achieve a minimum threshold of sensitivity (≥ 90%) is vital in clinical applications to reduce the risk of false negatives, which could lead to missed seizures and delayed interventions. In our experiments, the proposed Deepwalk-TS model achieves a sensitivity of 99.07%, which meets the necessity for ensuring timely detection of seizures. Meanwhile, high specificity (≥ 95%) is also important to avoid false alarms leading to unnecessary treatments or disruptions. Our model achieves a specificity of 98.97%, which is suitable for distinguishing between seizure events and non-seizure events, minimizing the occurrence of false positives. In addition, the overall accuracy (≥ 95%) and Area Under the Curve (AUC ≥ 0.98) are also crucial for evaluating the overall performance of the detection system. Our model achieves an accuracy of 99.54% and an AUC of 99.06%, exceeding the necessary thresholds for precise epilepsy detection. In a real-world deployment beyond performance metrics, real-time detection with low-latency processing is essential for timely seizure detection and intervention. Although our model achieves high accuracy and sensitivity, further optimization is required for efficient real-time operation in portable EEG devices or closed-loop systems.

6.How the proposed emotion classification system be useful in the real-world scenario? Need to discuss.

We understand the question asks how our seizure detection work can be applied to emotion classification in real-world scenarios. While our research primarily focuses on EEG-based seizure detection, the core module of Deepwalk-TS can capture both spatial and temporal features from raw EEG signals and can be effectively applied to other domains, including emotion classification.

We have included a discussion on the application of the Deepwalk-TS algorithm for emotion classification in the revised version of the Discussion section. Adapting real-time EEG processing to detect emotional states offers vast potential across healthcare, human-computer interaction, education, and more. Emotion detection using EEGs can help track emotional distress, such as anxiety or depression, aiding clinicians in identifying early signs for timely intervention. It can also enhance customer interactions by detecting emotions like frustration or satisfaction. Additionally, seizure detection methods could extend to entertainment, such as video games, by tailoring user experiences based on emotional states. By extending our work into emotion classification, future research could create systems that respond dynamically to a person’s emotional state, offering better decision-making in various real-world scenarios. We have expanded on the potential applications of emotion classification using similar techniques in the revised manuscript.

Reviewer #1: The authors have performed a study on a deep-learning based model to detect epileptic seizures. The paper is well-written and the proposed methodology is robust. Some minor points need to be revised.

Thank you very much for your positive feedback and constructive comments. We greatly appreciate your recognition of the value of our work and the robustness of methodology. We have carefully considered your comments and incorporated the necessary revisions to further improve the clarity and depth of our paper.

1.Introduction: - page 1 the authors state that “early diagnosis of seizures is important for patients with refractory epilepsy to prevent irreversible damage to the patient’s brain”. This sentence makes too many assumptions in implying that single epileptic seizures can damage the brain irreversibly. Seizure detection is an important goal, but this sentence needs to be reformulated.

We have revised the sentence and refined the logic of the introduction to ensure clarity and precision. The revised sentence now reads: "Early detection of epilepsy is crucial for timely intervention, reducing the risk of irreversible brain damage, and improving the overall quality of life for patients."

In this revised version, we emphasize the importance of early diagnosis in the high prevale

---

## [Decision Letter · Decision Letter 1]

28 Feb 2025

PONE-D-24-53519R1A Novel Dual-Branch Network for Comprehensive Spatiotemporal Information Integration for EEG-based Epileptic Seizure DetectionPLOS ONE

Dear Dr. Deng,

Thank you for submitting your manuscript to PLOS ONE. After careful consideration, we feel that it has merit but does not fully meet PLOS ONE’s publication criteria as it currently stands. Therefore, we invite you to submit a revised version of the manuscript that addresses the points raised during the review process.

We look forward to receiving your revised manuscript.

Kind regards,

Yuvaraj Rajamanickam, Ph.D

Academic Editor

PLOS ONE

**Journal Requirements:**

Reviewers' comments:

Reviewer's Responses to Questions

**Comments to the Author**

1. If the authors have adequately addressed your comments raised in a previous round of review and you feel that this manuscript is now acceptable for publication, you may indicate that here to bypass the “Comments to the Author” section, enter your conflict of interest statement in the “Confidential to Editor” section, and submit your "Accept" recommendation.

Reviewer #1: All comments have been addressed

Reviewer #2: All comments have been addressed

Reviewer #3: All comments have been addressed

2. Is the manuscript technically sound, and do the data support the conclusions?

Reviewer #1: Yes

Reviewer #2: Yes

Reviewer #3: Yes

3. Has the statistical analysis been performed appropriately and rigorously? 

Reviewer #1: Yes

Reviewer #2: Yes

Reviewer #3: Yes

4. Have the authors made all data underlying the findings in their manuscript fully available?

Reviewer #1: Yes

Reviewer #2: Yes

Reviewer #3: Yes

5. Is the manuscript presented in an intelligible fashion and written in standard English?

Reviewer #1: Yes

Reviewer #2: Yes

Reviewer #3: Yes

6. Review Comments to the Author

**Reviewer #1: **Thank you for addressing my previous comments and revising the manuscript accordingly. I appreciate the efforts made to improve the clarity and quality of the paper. The modifications have significantly enhanced the manuscript, and I find the current version to be well-structured and scientifically sound. I have no further major concerns at this stage.

**Reviewer #2:** Please find the attached file for the comments. The paper needs minor revisions before the next submission.

**Reviewer #3: **The author revised the manuscript where, all comments are addressed and responses are satisfactory. I do not have any further queries.

7. PLOS authors have the option to publish the peer review history of their article (what does this mean?). If published, this will include your full peer review and any attached files.

Reviewer #1: **Yes: **Umberto Aguglia

Reviewer #2: No

Reviewer #3: No

---

## [Author Response · Author response to Decision Letter 2]

11 Mar 2025

Response to Reviewers

Dear Editor and Reviewers:

We sincerely appreciate your letter and the reviewers’ valuable feedback on our manuscript titled "A Novel Dual-Branch Network for Comprehensive Spatiotemporal Information Integration for EEG-based Epileptic Seizure Detection" (PONE-D-24-53519R1). We are grateful for the reviewers’ positive comments, which have helped us further refine and improve our work. We have carefully reviewed all the suggestions and have made the necessary revisions in the Minor Revision resubmission. Below are our detailed point-by-point responses to each comment.

Editor Comments:

1.Please review your reference list to ensure that it is complete and correct.

We have carefully reviewed our reference list and thoroughly checked each citation to ensure completeness and accuracy. Missing fields, such as page numbers and other relevant details, have been corrected and supplemented where necessary. The revised manuscript now includes a fully verified and properly formatted reference list that adheres to the journal’s guidelines.

2.While revising your submission, please upload your figure files to the Preflight Analysis and Conversion Engine (PACE) digital diagnostic tool.

Thank you for your valuable feedback, which has helped us improve the overall quality of our submission. We have carefully reviewed all figure files to ensure they meet the journal's formatting requirements. Please let us know if any further modifications are needed. We appreciate your guidance and support throughout the revision process.

Reviewer #1: The modifications have significantly enhanced the manuscript, and I find the current version to be well-structured and scientifically sound. I have no further major concerns at this stage.

Thank you for your positive feedback and recognition of our efforts in improving the manuscript. We sincerely appreciate your thoughtful review and valuable suggestions, which have helped us enhance the clarity, structure, and scientific rigor of our work. Your support and encouragement are greatly appreciated!

Reviewer #2: Please find the attached file for the comments. The paper needs minor revisions before the next submission.

1.The Fig.6 results show the overfitting issue. Please check it. The validation accuracy is higher than the training accuracy in the case of the SEED dataset.

We sincerely thank the reviewer for his insightful comments on the potential overfitting issue in the loss curve. Generally, a larger gap between the training and validation curves indicates strong overfitting, while a minimal gap indicates less severe overfitting. After carefully re-evaluating the training and validation process, we found that the original validation curve was not fully plotted, which could have led to a misinterpretation of the results. We have corrected this issue in the revised figure and observe that the validation accuracy remains consistent with the training accuracy, indicating that the overfitting is small and does not significantly affect the overall model performance.

2.Please consider the following reference for your discussion

We sincerely appreciate the reviewer’s suggestion to incorporate this reference into our discussion. The recommended paper offers valuable insights into EEG-based emotion recognition and the comparative performance of various supervised machine learning algorithms. We have now cited this reference in the revised manuscript, and we greatly appreciate the reviewer’s insightful recommendation.

Reviewer #3: The author revised the manuscript where, all comments are addressed and responses are satisfactory. I do not have any further queries.

Thank you for your thoughtful review and positive feedback. We sincerely appreciate your time and effort in evaluating our manuscript. Your valuable comments have helped us improve the quality of our work, and we are grateful for your support!

We greatly appreciate the positive feedback regarding the potential and value of our work. We are committed to addressing these concerns and will revise the manuscript to improve its clarity and completeness. We are excited to submit the revised version for you to look over. If you have any queries, please don’t hesitate to contact me at the e-mail below.

Sincerely,

Xiaobing Deng

guoguo10201@163.com

---

## [Editor Report · Decision Letter 2]

14 Mar 2025

A Novel Dual-Branch Network for Comprehensive Spatiotemporal Information Integration for EEG-based Epileptic Seizure Detection

PONE-D-24-53519R2

Dear Dr. Deng,

We’re pleased to inform you that your manuscript has been judged scientifically suitable for publication and will be formally accepted for publication once it meets all outstanding technical requirements.

Kind regards,

Yuvaraj Rajamanickam, Ph.D

Academic Editor

PLOS ONE
---

## [Editor Report · Acceptance letter]

PONE-D-24-53519R2

PLOS ONE

Dear Dr. Deng,

I'm pleased to inform you that your manuscript has been deemed suitable for publication in PLOS ONE. Congratulations! Your manuscript is now being handed over to our production team.

Kind regards,

on behalf of

Dr. Yuvaraj Rajamanickam

Academic Editor

PLOS ONE